# *Aphidius colemani* Behavior Changes Depending on Volatile Organic Compounds Emitted by Plants Infected with Viruses with Different Modes of Transmission

**DOI:** 10.3390/insects15020092

**Published:** 2024-01-29

**Authors:** Gemma Clemente-Orta, Ángel Cabello, Elisa Garzo, Aranzazu Moreno, Alberto Fereres

**Affiliations:** 1Instituto de Ciencias Agrarias, Consejo Superior de Investigaciones Científicas (ICA-CSIC), C/Serrano 115 dpdo, 28006 Madrid, Spain; angelcabelloberzosa@gmail.com (Á.C.); amoreno@ica.csic.es (A.M.); afereres@ica.csic.es (A.F.); 2Departament de Producció Vegetal i Ciència Forestal, AGROTECNIO Center, Universitat de Lleida, Rovira Roure 191, 25198 Lleida, Spain

**Keywords:** biological control, aphid, parasitoids, multitrophic interactions, single and mixed viral infection

## Abstract

**Simple Summary:**

Plant viruses and other pathogens manipulate their hosts to alter the attractiveness and palatability of their insect vectors, thereby enhancing their own spread. For instance, melon plants infected with cucumber mosaic virus (CMV), a non-persistent virus, release specific volatile organic compounds (VOCs) that attract their primary vector, *Aphis gossypii*. These volatiles, emitted after aphid colonization, often act as a “cry for help” signal, attracting natural enemies. Surprisingly, studies have not delved into how aphid parasitoids respond to VOCs emitted by plants infected with viruses with different modes of transmission: non-persistent, persistent, or a combination of both. In our investigation, we explored how single and mixed infections with CMV (non-persistent virus) or/and cucurbit aphid-borne yellowing virus (CABYV, persistent virus) modify the attractiveness of *Aphidius colemani* to aphid-free and aphid-infested melon plants. These experiments highlight the need for further investigation into the complexities of parasitoid behavior, providing insights into the long evolutionary processes shaping plant–aphid–parasitoid interactions and their impact on virus transmission dynamics.

**Abstract:**

Natural enemies are an additional component that may interact directly with the plant–virus–vector association, affecting viral dispersion. In our study, we conducted olfactometry assays to explore how single and mixed infections with CMV or/and CABYV modify the attractiveness of *A. colemani* to aphid-free and aphid-infested melon plants using two melon genotypes. Subsequently, we investigated the influence of CABYV-infected plants infested by *A. gossypii* on the parasitism rate and emergence of *A. colemani* in a dual-choice assay under greenhouse conditions. Our study demonstrates that males showed no preference for either infected or non-infected plants. Female parasitoids exhibit a preference for volatiles emitted by CMV and mixed-infected melon plants over clean air but not over mock-inoculated plants, suggesting a response influenced by plant genotype. Female parasitoid responses to CABYV and its interactions with aphids revealed a preference for mock-inoculated plants over CABYV-infected plants and a parasitism rate slightly higher (7.12%) on non-infected plants. Our study revealed that (1) parasitoids may reject olfactory cues from CABYV-infected plants, potentially interfering with the plant’s “cry for help” response; (2) in the case of CMV, whether in single or mixed infections, non-infected plants are as attractive as infected ones to parasitoids. Our findings suggest that persistent viruses manipulate aphid parasitoid behavior to their advantage, promoting virus disease in melon crops.

## 1. Introduction

Melon (*Cucumis melo*) holds the position of the third most important horticultural crop in Spain, following tomato and watermelon. This significant production has made Spain the leading exporter of melons globally, with a substantial portion of exports directed to countries, such as France and Germany, within the European Union [1]. Melons in Spain are frequently cultivated under integrated pest management (IPM) programs, which play a crucial role in sustainable farming practices. Although various insects pose threats to melon crops by diverting essential plant nutrients through feeding, insect-transmitted viruses are particularly noteworthy. Aphids, among the most economically significant pests in temperate agriculture, cause substantial damage to plants through both direct feeding and, more importantly, as vectors for viruses [2]. Plant viruses commonly rely on an arthropod partner, known as a “vector”, for efficient transmission to new hosts [3]. Viral infections can induce changes in host plants, resulting in indirect effects such as the emission of volatile organic compounds (VOCs), alterations in plant nutrients, and changes in plant morphology [4,5]. These alterations often render the host plant more attractive and suitable for the vector [6]. Furthermore, the direct effects of viruses on the insect vector, including changes in vector biology, fitness, and behavior, significantly impact virus transmission and disease dynamics [7].

*Aphis gossypii* (Hemiptera: Aphididae) is the primary insect vector in melon crops. Two of the most important viruses present in melons are cucumber mosaic virus (CMV, Cucumovirus), transmitted in a non-persistent manner, and cucurbit aphid-borne yellow virus (CABYV, Polerovirus), transmitted in a persistent circulative manner. Previous studies have indicated that CMV-infected plants are more attractive to aphid landing, although they are not preferred for feeding soon after aphid probing—a behavior known to enhance the transmission efficiency of non-persistent viruses [8]. Furthermore, CABYV induces changes in its host plant that modify aphid settlement and feeding behavior, promoting long-term feeding and enhancing virus acquisition from infected plants. Once aphids become viruliferous, they prefer to settle and feed on healthy plants, optimizing the transmission and spread of this phloem-limited virus [9]. These results suggest that persistent viruses can manipulate the behavior of their aphid vector to enhance transmission. Both CMV and CABYV can be found in both single and mixed infections in melon plants. Each of these pathogens has economic impacts on melon crops, with yield losses reaching up to 50% for CMV [10] and up to 100% for CABYV [11,12]. Mixed infections are common in melon crops in Southeastern Spain, where CABYV was the most prevalent virus in recent years [13]. Yield losses can increase significantly when both viruses co-infect melon plants. However, the aphid response to plants infected with both viruses remains unknown.

The host plant, the insect vector, and the plant virus constitute interdependent components of a complex pathosystem [14], resulting in a three-way plant–virus–vector association termed a “tripartite” relationship. Building on the concept of viral infections, it is noteworthy that when phytophagous insects feed on plants, the emission of volatile organic compounds occurs, known as herbivore-induced plant volatiles (HIPVs), which may also attract natural enemies [15]. This raises the possibility that natural enemies could be an additional component that directly interacts with the plant–virus–vector tripartite association. These interactions may have an impact on plant virus transmission, leading to a transition from a tripartite to a multitrophic interaction association. In light of this information, a logical expectation would be to anticipate a decrease in vector numbers as a consequence of natural enemy activity, subsequently resulting in reductions in virus transmission [6,16].

In multitrophic interactions, three signals of communication have been proposed to elucidate the influence of natural enemies on the spread of insect vectors and vector-borne viruses. These signals include the following: (1) The alarm signal or alarm pheromone (kairomone) (aphids, acting as vectors, emit an alarm signal, also known as an alarm pheromone, to alert their conspecific community). (2) Vector–host plant preference (the “vector–host plant preference” serves as a strategy employed by the virus to enhance the fitness of the vector. This preference can lead to changes in vector morphology, physiology, or behavior, ultimately impacting virus spread). (3) Cry for help signal (Plants, when under herbivore attack, emit a “cry for help signal”. This mechanism is designed to attract natural enemies, contributing to the plant’s defense against herbivores) [6,17,18]

The recent literature underscores the intricate dynamics between natural enemies, insect vectors, and viruses, revealing that the impact of natural enemies on their insect vector prey depends on factors such as the level of disturbance imposed by the prey and the mode of virus transmission by the vector. The level of disturbance in the spread of vector-borne viruses is intricately linked to the consumptive and non-consumptive activities of predators and parasitoids. These activities can induce changes in vector morphology, physiology, or behavior. For instance, promoting the development of winged aphids may facilitate virus spread [19]. Laboratory studies in controlled conditions have demonstrated varied effects on virus spread. Ladybeetles (*Adalia bipunctata*) led to a higher increase in the spread of broad bean wilt virus 1 (BBWV-1, Fabavirus) compared to syrphids *(Sphaerophoria rueppellii*) on pepper plants [20]. Moreover, the presence of both the predator (*Coccinella septempunctata*) and the parasitoid (*Aphidius rhopalosiphi*) resulted in differential virus spread over time. Durum wheat seedlings infected with the persistent transmitted virus barley yellow dwarf virus (BYDV, Luteovirus) showed more extensive infection in the presence of parasitoids than in control plants [21].

The influence of parasitoids on virus spread varies depending on the mode of transmission. For example, the bird cherry-oat aphid *Rhopalosiphum padi*, a vector of persistent virus (CYDV, Luteovirus) in wheat plants, exhibited reduced attacks by *Aphidius colemani* when actively harboring the plant pathogen [22]. *A. colemani*’s response in different conditions, for example, in the greenhouse, promoted early dispersal of aphids, leading to an increased incidence of non-persistent virus CMV in the short term (2 days). However, the consequences of parasitism suggested potential long-term benefits. Additionally, *A. colemani* significantly limited the spread of the persistent virus CABYV in the long term (14 days) [23]. In semi-natural foraging assays, higher rates of aphid parasitism by *A. colemani* were observed on CMV-infected plants compared to healthy plants. This difference was attributed to the successful development of parasitoid larvae when aphid hosts fed on CMV-infected plants [24]. Similarly, in the presence of the persistently transmitted turnip yellows virus (TuYV, Polerovirus) infecting bastard saffron plants, *A. colemani* exhibited a similar response to infected and uninfected plants. However, parasitoids emerged smaller when developing from viruliferous aphids compared to those from non-viruliferous aphids [25].

Regarding the attraction to infected plants, in the absence of their vector host, some studies have revealed the attraction of parasitoids to plants infected with various viruses, including tomato yellow leaf curl virus (TYLCV, Geminivirus) [26] and the bacteria causing huanglongbing (HLB) disease [27]. Additionally, recent findings have indicated that pepper plants infected with non-persistent viruses (CMV and Potato Virus Y, PYV, Potyvirus) emitted volatiles detected by the parasitoid *A. colemani* [28]. Moreover, the volatile compounds induced by rice dwarf virus (RDV, Reovirus) infection attract the natural enemy *Cyrtorhinus lividipennis* to prey on its virus vectors [29].

Despite the numerous studies investigating how vector behavior can influence virus spread, our understanding of the role of natural enemies in this complex multitrophic interaction remains incomplete. While the olfactory response of parasitoids to plants infected with non-persistent viruses in single infections has been documented, there is a notable gap in our knowledge regarding parasitoid responses to plants infected by multiple viruses, especially in the presence or absence of their vector with different transmission modes. Biological control programs for aphid-borne viruses heavily rely on parasitic wasps. However, these programs often overlook how parasitoids and parasitism rates may be affected in the presence of virus-infected plants. As we continue to refine strategies for biological control, investigating the nuanced interactions between parasitoids, vectors, and viruses in multitrophic systems becomes imperative for more effective pest management. Building on this background, in our virus–aphid–plant interaction study, we focused on *A. colemani,* a polyphagous aphid parasitoid originally from Northern India or Pakistan, which has expanded its habitat to North and South America, Australia, and Europe [30]. This parasitoid species is recognized for its role in aphid control and exhibits an innate preference for *A. gossypii* [31].

The current study aimed to provide new insights into how melon plants infected with viruses with different modes of transmission—CMV and CABYV—may affect the behavior of *A. colemani* in the absence or presence of its aphid host, *A. gossypii*. Thus, we investigated the specificity of these responses when the two viruses were present in single and mixed infections, using two melon genotypes with different degrees of tolerance to CABYV. Our study included (1) olfactometry bioassays and (2) a dual-choice bioassay to evaluate the parasitism and emergence rate of *A. colemani* on CABYV-infected and mock-inoculated plants.

## 2. Materials and Methods

### 2.1. Plant Growing Conditions

We employed two *Cucumis melo* genotypes in our study. The first genotype, cultivar Bazán (Semillas Fitó S.A., Barcelona, Spain), was selected for its tolerance to CABYV, as confirmed by virus symptoms observed in previous lab assays [32]. The second genotype, accession C311 (IHSM-La Mayora, Málaga, Spain), served as a CABYV-sensitive host [33]. Both genotypes showed CMV mosaic symptoms. Plants were transplanted into 11 × 11 × 12 cm pots at 7 days old (cotyledon stage) using a substrate mixture of 1:2 vermiculite (No. 3, Asfaltex S.A., Barcelona, Spain) and soil substrate (GoV4, Jiffy International, A.S., Stange, Norway). The plants were maintained inside a plant growth chamber under controlled conditions: 24:20 °C (Day:Night) temperature, 60:80% relative humidity (RH), and a 16:8 h (Light:Dark) photoperiod. Regular watering and bi-weekly fertilization (1 g/L, Miller Chemical & Fertilizer Corp., PA, USA) were administered to ensure optimal plant growth.

### 2.2. Insect Rearing

Laboratory colonies of non-viruliferous *A. gossypii* were reared on melon plants (cv. Bazán) within rearing cages placed in environmental growth chambers. The growth chambers maintained conditions of 23:18 °C (Day:Night) temperature, 60–80% relative humidity (RH), and a 14:10 h (Light:Dark) photoperiod. Mummies of *A. colemani* were supplied by Koppert Biological Systems (Berkel en Rodenrijs, the Netherlands). Prior to the trials, parasitoids underwent at least three generations of rearing on *A. gossypii* feeding on melon plants (cv. Bazán) in a growth chamber with conditions set at 25:20 °C (Day:Night) temperature, 60–80% RH, and a 16:8 h (Light:Dark) photoperiod. Parasitoid density in the main rearing (mass breeding of parasitoids) was restricted to avoid a male-biased sex ratio, a precursor to population decline.

For synchronized rearing, small colonies (approximately 10–15 adults) of *A. gossypii* were initiated in individualized plastic cylinders with a melon plant in the same growth chamber. After 48 h, all adult aphids were removed. When 3rd instar aphid nymphs were present on the melon plants, parasitoids (approximately 10 adults with a sex ratio of 5:5) were released. One-to-four-day-old males and females of *A. colemani* were collected from the plastic cylinders using an insect aspirator. Each parasitoid was individually placed in a plastic vial (diameter 2.5 cm, height 5 cm) stoppered with agrotextile, with water and a honey drop provided as a food source. Adults for the olfactometry bioassays were collected at least 2 h before the experiment, individually sexed, and prepared for the experiments.

### 2.3. Virus Isolates and Inoculations

CMV-infected plants underwent mechanical inoculation with a CMV isolate (isolate M6) obtained from a melon crop in Tarragona, Spain, in 1996, provided by Dr. E. Moriones (IHSM-La Mayora, Málaga, Spain). Inoculation occurred two weeks after sowing at the 1 true leaf stage (102 BBCH; BBCH scale [34]). Mock-inoculated cucumber plants, rubbed only with a specific buffer solution and carborundum, at the same growth stage, served as controls. Visual inspection for mosaic symptoms on leaves and a Double Antibody Sandwich Enzyme-Linked Immunosorbent Assay (DAS-ELISA) [35] using specific antibodies (Bioreba A.G., Reinach, Switzerland) confirmed CMV infection before olfactometry bioassays.

CABYV-infected plants were inoculated with a CABYV isolate provided by Dr. H. Lecoq, sourced from zucchini squash in Montfavet, France, in 2003. The isolate was propagated and maintained in cucumber plants at ICA-CSIC through aphid serial transmission. Inoculation of cucumber plants, performed two weeks after sowing at the 1 true leaf stage (102 BBCH), involved 20 viruliferous nymphs reared on CABYV-infected plants. Aphids were confined using insect clip cages during a 72 h inoculation access period (IAP) and then removed. Non-infected cucumber plants at the same growth stage were inoculated with 20 non-viruliferous nymphs and used as mock-inoculated control plants. DAS-ELISA with specific antibodies (Sediag S.A.S., Bretenière France) confirmed CABYV infection before olfactometry bioassays.

Mixed-infected plants underwent initial CABYV inoculation by aphids using the aforementioned protocol. Subsequently, an opposite leaf on the same plant was mechanically inoculated with CMV. Mock-inoculated control plants at the same growth stage were inoculated with 20 non-viruliferous nymphs for 72 h, after which all nymphs were removed. Mechanical rubbing with specific buffer solution and carborundum followed. DAS-ELISA confirmed single and mixed infections, and mock-inoculated plants were used for all bioassays 4 to 5 weeks post-inoculation (or 50 days post-sowing) when viral infections were confirmed. Melon plant photos illustrating mock and infected plants with viral symptoms were recorded in all treatments (Appendix A).

### 2.4. Aphid Infestation of Viral-Infected Plants

At four weeks post-inoculation, melon plants infected with CMV, CABYV, or a mixture of both (mixed-infected) were infested with one hundred 2nd- to 3rd-instar nymphs of *A. gossypii*. All aphids remained on both mock-inoculated and virus-infected plants for a total of 48 h. This ensured that the infesting aphids reached the 4th nymphal instar, providing uniformity in aphid age for subsequent olfactometer bioassays, following a procedure previously described by [36]. Following the infestation period, all melon plants were maintained in a growth chamber at 25:20 °C (Day:Night) temperature, 60–80% relative humidity (RH), and a 16:8 h (Light:Dark) photoperiod during the two days of olfactometer bioassays. Subsequently, the plants were discarded to prevent aphids from reaching the adult stage.

### 2.5. Olfactometer Bioassays

The bioassays were conducted using a two-way olfactometer composed of a VidraFOC glass Y-tube with an inner diameter of 18 mm, a 90 mm stem, and two arms of 100 mm assembled at an angle of 120° (VidraFOC, S. A, Barcelona, Spain). Each arm was connected to a 5 L Closed Volatile Collection Chamber (VCC) of VidraFoc containing a plant as the odor source. The Y-tube was connected through three male glass adapters, with two of them linked to a Teflon tubing extruded (Straight) (TUB-1/4-TFE-EX), controlling airflow using a flow meter in an Air Delivery System (ADS) (OLFM-2C-ADS+B model 2510A2A29BNBN, Brooks Instrument, Hatfield, PA, USA). The ADS was attached to a 3-stage air filtration system. An airflow of 0.7 L/min (previously determined as the optimal flow for parasitoid response) passed through each olfactometer arm (Figure 1). Additionally, a flowmeter at the end of the Teflon tube of each VCC chamber confirmed the correct airflow reaching the Y-tube.

Bioassays were conducted on a lab bench under ambient environmental conditions (24 ± 2 °C, 50 ± 20% RH) from 08:30 a.m. to 4:30 p.m., illuminated with daylight LED lamps. Melon plants were placed in the VCC chamber for 30 min without airflow and an additional 30 min with airflow before each bioassay. Each parasitoid released into the Y-tube had a maximum of 10 min to make a choice. Parasitoids were considered to have made a choice when positioned 70 mm above the bifurcation for at least 2 min (Figure 1). Parasitoids that did not make a choice within the specified time were recorded as non-choice and excluded from the dual-choice statistical analysis. Both non-choice and choice data were used to analyze adult behavior. Each round consisted of 10 individuals of *A. colemani* and 2 pairs of plants per repetition using 1 pair of different odor sources (10 replicates per repetition). The Y-tube glass was replaced every five insects, and the positions of the odor sources were interchanged for each parasitoid to prevent bias in choice. Odor source treatments were renewed every 10 insects, and all glassware was washed with 2% soapy water, purified water, 90° ethanol, 99% acetone, and then dried in a fume hood for at least 24 h. Combinations of treatments were tested using both cv. Bazán and C311 melon plants with experienced male and female individuals (Appendix A). An additional set of bioassays was conducted using virus-infected plants previously infested by *A. gossypii* nymphs, as described above, using both melon genotypes and experienced females.

### 2.6. Dual-Choice Bioassay on the Parasitism Rate of Aphidius colemani on CABYV-Infected and Mock-Inoculated Plants

A semi-field experiment was conducted in the greenhouse facilities at ICA-CSIC to evaluate the parasitism and emergence rates of *A. colemani* exposed to CABYV-infected or mock-inoculated melon plants previously infested by aphids. Each plant was infested with 30 adults of *A. gossypii* of the same age. To synchronize aphids, adults were removed from clip-cages after a 24 h infestation period, and newborn nymphs were allowed an additional 48 h period until they reached the 2nd–3rd nymphal instar. At least one hundred nymphs on each infested plant were present before the experiment started. One-to-two-day-old males and females of *A. colemani* were collected using an insect aspirator, individualized in plastic vials for at least 2 h, and sexed before the bioassays started. Finally, 16 females and 4 males were placed together in a plastic vial (12 × 22 cm) with water and a honey drop as a food source for 24 h before releasing them into the cages. The parasitoids were released in the middle of an aphid-proof large mesh cage (1 m^3^), where the two test plants were placed on opposite sides (both plants were separated by 80 cm). In each round, three large mesh cages were used separated by 40 cm between them (four replicates in total). Each cage contained aphid-infested plants (aphids in the 4th nymphal instar), one infected with CABYV on one side and a mock-inoculated on the opposite side. Parasitoids were allowed to forage on aphids for 24 h and then manually removed from the cages. The parasitism and emergence rates were assessed 11 days after parasitoid release using destructive counting on the leaves where one hundred nymphs were released. Temperature, relative humidity, and shadow covers were remotely controlled in the greenhouse through a central computer to ensure the following environmental conditions: a temperature of 25:20 °C (Day:Night), a photoperiod of 16:8 h (Light:Dark), and 75–80% RH.

### 2.7. Statistical Analysis

In the Y-tube olfactometer bioassay, data were analyzed using a Chi-squared test (*p* < 0.05) to determine whether volatiles emitted by infected plants, compared to mock-inoculated or clean air, had a significant effect on parasitoids. For the dual-choice host-seeking behavior assay, data were analyzed using a Student’s *t*-test assuming normal distribution (*p* < 0.05). The statistical analyses were performed using R version 4.2.2.

## 3. Results

During the olfactometry assays, a total of 1622 *A. colemani* individuals were tested, including 1114 females and 508 males, using 95 melon plants. In the dual-choice host-seeking behavior assay, a total of 320 *A. colemani* individuals were tested, comprising 256 females and 64 males, using 24 melon plants.

### 3.1. Effect of Virus-Infected Melon on the Olfactory Response of Aphidius colemani

The results indicate that *A. colemani* males did not show a preference for healthy melon plants over clean air or mock-inoculated melon over CMV-infected, CABYV-infected, or mixed-infected melon plants across both genotypes (Appendix A). The only exception was observed in the case of C311 (which is sensitive to CABYV), where males preferred clean air over CMV-infected melon plants (χ^2^ = 5.76; df = 1; *p* = 0.016). Additionally, when healthy melon plants were infested by *A. gossypii* in the control treatment, males showed no preference for either healthy melon over clean air in cv. Bazán (χ^2^ = 1.2; df = 1; *p* = 0.273) and C311 (χ^2^ = 1.12; df = 1; *p* = 0.288). No further treatments were tested with males after this observation.

In Figure 2, it can be seen that *A. colemani* females respond to the volatile cues from virus-infected melon plants, considering different genotypes and virus infections. *A. colemani* females exhibited preferences for volatiles emitted by CMV-infected and mixed-infected melon plants over clean air in cv. Bazán (which is tolerant to CABYV). However, this preference was not observed when infected plants were compared to mock-inoculated plants (Figure 2a). On the other hand, in the case of C311 (which is sensitive to CABYV), females showed no preference for healthy melon over clean air, and they did not exhibit preferences for CMV, CABYV, or mixed-infected melon over clean air or mock-inoculated melon (Figure 2b).

In Figure 3, it can be seen that *A. colemani* females respond to the volatile cues from virus-infected melon plants infested by *A. gossypii*, considering different genotypes and virus infections. Females showed no preference for clean air or mock-inoculated plants over CMV or mixed-infected melon plants when the plants were infested by *A. gossypii* (Figure 3a). In addition, females preferred CMV-infected and mixed-infected melons over clean air melons, but this response was not found when infected plants were compared to mock-inoculated plants (Figure 3b).

Moreover, females exhibited a preference for mock-inoculated plants over CABYV-infected melon in both genotypes infested by *A. gossypii* (sensitive and tolerant to CABYV). C311 showed a slightly higher proportion of females preferring mock-inoculated plants compared to cv. Bazán. In C311, females showed a statistically significant preference for clean air over CABYV-infected melons (Figure 3b), while in cv. Bazán, the preference for clean air over CABYV-infected melons was statistically non-significant (Figure 3a). Comparing CABYV-infected plants also infested by *A. gossypii* between genotypes, parasitoid no response to volatiles emitted by CABYV-infected plants (χ^2^ = 0.27; df = 1; *p* = 0. 6) (Appendix A) indicating no significant difference in the response to the two genotype in these case. Comparing CABYV-infected plants within each genotype, i.e., CABYV-infected melon over CABYV-infected melon also infested by *A. gossypii,* females did not show a preference for either C311 (χ^2^ = 0.03; df = 1; *p* = 0.85) or cv. Bazán (χ^2^ = 0.12; df = 1; *p* = 0.72) plants, indicating no significant difference in the response to these two treatments (Appendix A). 

### 3.2. Effect of CABYV-Infected Melon on the Parasitism Rate of Aphidius colemani

To further investigate parasitoid response to virus-infected plants and to confirm the result observed in the olfactory tests, where parasitoids were more attracted to mock-inoculated than to CABYV-infected plants, we performed a dual-choice bioassay in semi-field greenhouse conditions. The statistical analysis results showed no significant differences in parasitism rate (t = 0.7, df = 1, *p*-value = 0.49), but it was slightly higher on non-infected (44.92% ± 6.8) than on CABYV-infected plants (37.80% ± 7.46) (Figure 4a). Similarly, the emergence rate did not show significant differences between the treatments (t = 0.14, df = 1, *p*-value = 0.88) (Figure 4b). 

## 4. Discussion

Our study provides valuable insights into the complex interactions among plants, viruses, aphids, and parasitoids, shedding light on the intricate multitrophic relationships involved in the spread of vector-borne viruses. The inclusion of two virus species with different modes of transmission (non-persistent and persistent) and the examination of their effects on the behavior of parasitoids, especially in the presence or absence of their insect vectors, contribute to a comprehensive understanding of these interactions. The role of VOCs emitted by virus-infected plants in attracting or repelling parasitoids is a crucial aspect of this study. The observation that parasitoid responses vary depending on the virus species, transmission mechanism, and the tolerance of the plant to virus infection highlights the complexity of these interactions. The findings emphasize the need for a nuanced understanding of multitrophic interactions in the context of vector-borne viruses. Investigations on the role of VOC emitted by virus-infected plants and their insect vectors have focused mainly on how they affect disease spread, but the effect of this tritrophic relationship between the plant, the virus and the insect vector is unknown in the fourth trophic level, that is, the impact of plant-volatiles on the NE [6]. Recently, some studies have shown attraction of parasitoids to virus-infected plants in the absence of their insect vector [27,28,37]. Also, three studies have shown attraction of parasitoids to virus-infected plants in the presence of its vector hosts [24,29,37]. There is also a case where the plant pathogen did not modify parasitoid behavior [38]. This study contributes valuable information that can inform strategies for the biological control of aphid-borne viruses and enhance our understanding of the ecological dynamics at play in plant–virus–insect interactions.

### 4.1. Effects of Volatiles Emitted by Infected-Melon on Male Parasitoids

The inclusion of male *A. colemani* in our study adds an interesting dimension to understanding the potential impact of non-consumptive activities on virus spread. The observation that males did not show a preference for volatiles emitted by either non-infected or virus-infected plants, as well as non-infested or aphid-infested plants, suggests that their olfactory response in our experimental setup did not exhibit a discernible bias. For our consideration, the non-consumptive activities of males, such as actively searching for virgin females and influencing the behavior of non-receptive females, aligns with the broader ecological context in which these parasitoids operate [17,31,39,40,41]. Even though our olfactometry bioassays did not reveal a preference in male behavior regarding volatile emissions from infected plants, it is crucial to recognize that the field conditions and the dynamic nature of ecological interactions might lead to different outcomes. The complexity of these interactions underscores the importance of considering multiple factors, including the behavior of both males and females, in understanding the broader ecological implications of parasitoid–virus–plant interactions. It is possible that other cues or factors in the field could influence the behavior of male parasitoids and contribute to the overall dynamics of virus spread.

### 4.2. Effects of Volatiles Emitted by Infected Melon on Female Parasitoids in Absence and Presence of Its Main Aphid Host

Plants often provide the first cue in the chain of events that leads to the host location, regardless of the nature of the orienting factor [42]. Parasitoids seek an optimal host in a complex process that involves different types of signals (physical, chemical, or visual cues, orientation, alignment, or host quality), being fundamental for female parasitoids in the process of recognizing the host, evaluating and accepting (oviposition) or rejecting their prey in the event that it is not optimal [17]. Thus, behavioral studies have demonstrated the role of plant volatiles in foraging behavior, mainly by female parasitoids, and how they are affected by host phenology, blooming [36], plant biodiversity [43], or plant cultivar [44,45,46]. Recently, Milonas et al. [28] found that tomatoes infected by CMV produce plant volatiles that are detected by *A. colemani* in the absence of its aphid host. Following this result, we found, in our study, that females of *A. colemani* showed a preference for volatiles emitted by CMV and mixed-infected melon plants when compared to air in cv. Bazán (tolerant to CABYV) in the absence of their aphid host. In the case of the other genotype used, C311 (sensitive to CABYV), the responses of females followed the same tendency but were not statistically significant. The preference of parasitoids for odors emitted by mixed-infected plants has not been previously reported. These findings are not a consequence or causality of parasitoids detecting plants over clean air because this did not happen in the rest of the treatments. Such evidence could suggest that in the host selection behavior of parasitoids, the volatile cues derived from infected plants, by a non-persistent virus, could have a similar influence as signals emanating directly from a non-infected host plant.

Thus, our results agree with previously reported findings regarding CMV attracting parasitoids to plant infections [28], but we add to this knowledge that it could happen when CMV is in single and mixed infections. Regarding the genotype effect, when aphids were feeding on virus-infected plants, female parasitoids showed a preference for odors emitted by CMV and mixed-infected plants (in the case of the C311 genotype but not for cv. Bazán), suggesting that volatile emissions may vary depending on the plant genotype that is infected with a plant virus. It is known that the effects of CMV change the phenotype of the host plant (cucumber) that influences plant–aphid interactions, reducing host plant quality for aphids, causing rapid vector dispersal, and enhancing virus spread [47]. Also, no significant differences were found in parasitoid attraction to healthy over CMV-infected plants infested with *Myzus persicae* in Y-tube olfactometer assays [24]. These results suggest that the increased parasitism rates observed in previous studies were not explained by the enhanced attraction of parasitoids to the odors of infected plants but were a result of CMV-induced changes in host plant nutrition, reducing aphid diet during parasitoid development and compromising their ability to mount effective defenses against parasitism. The viral infection of plants interferes with physiological volatile defense signaling, creating more favorable conditions for *A. gossypii*, but aphid infestation may also induce the emission of plant terpenoids and, in some cases, reductions in other VOCs that can attract aphid parasitoids to aphid-infested plants [48,49]. Following these results, we did not find significant differences between the selection of parasitoids of mock-inoculated over CMV-infected plants in the presence of its aphid vector as shown in previous studies [24,47], but we found significant differences between CMV-infected and clean air plants.

Despite hardly any response being found to volatiles emitted by mixed-infected plants, it is known that plant viruses can be found either in a single or mixed infection on melons, and this has a direct impact on the plant host and also potentially on their relationships with their insect vector compared to simple infection [50,51]. These double infections often result in more severe visual symptoms than single infections of the same viruses, increasing vector attraction via visual cues [52]. Therefore, studies regarding mixed infections’ effects on their vector must be considered in further investigations.

Recently, Roudine et al. [19] found that a total of 11 studies were published regarding the relationship between NE and circulative virus incidence: 5 showed a negative effect, 3 revealed a positive effect, and 2 studies had opposite results, depending on the duration of the trial and the mode of virus transmission [23,52]. They concluded that, in general, the anti-predator behavior of insect vectors, such as higher movement between host plants, does not seem to be a key determinant of the effects of natural enemies on persistently transmitted virus spread [19]. Regarding persistent viruses, there is no information to date regarding the effects of volatiles emitted by CABYV-infected plants on predators or parasitoids. Until now, a different approach has been used to study how parasitoids could influence CABYV spread in the presence of aphid vectors [23]. They reported that *A. colemani* significantly limited the spread and incidence of CABYV by *A. gossypii* in the long term in greenhouse conditions. In our study, our approach was different; we wanted to study how parasitoids reacted to volatiles emitted by CABYV-infected plants in the absence or presence of aphids. We found that *A. colemani* showed no preference for volatiles of CABYV-infected plants in the absence of aphids, but female parasitoids showed a clear preference for mock-inoculated over CABYV-infected plants in both melon cultivars. Thus, our results suggest that parasitoids could reject olfactory cues emitted by CABYV-infected melon plants, possibly interfering with the “cry for help” plant response that has been reported to attract parasitoids after aphid colonization. To elucidate if the response of parasitoids to CABYV-infected plants was related to the presence of aphids in the infected source plant, we ran an extra parasitoid choice test between volatiles emitted by CABYV-infected plants over CABYV-infected plants also infested by *A. gossypii.* The results showed no statistically significant differences between treatments for either of the two melon genotypes evaluated. Consequently, to further investigate parasitoid attraction to mock-inoculated over CABYV-infected plants, we conducted a semi-field experiment to test if the parasitoids really preferred to parasitize aphids on mock-inoculated plants over aphids feeding on CABYV-infected plants. Our results did not show any significant differences between parasitism and emergency rates on infected CABYV over non-infected melon plants under greenhouse conditions. However, the parasitism rate was slightly higher (7.12%) on non-infected than on CABYV-infected plants. This difference between parasitism rates in CABYV and mock-inoculated plants could affect the efficacy of aphid biological control programs, but further studies are needed to test if parasitoids really reject CABYV-infected plants under field conditions. Other factors, such as visual cues, may play a more important role in host-seeking behavior. Parasitoids may be attracted to the yellowing symptoms of CABYV-infected plants, counteracting the non-preference response induced by VOCs observed in our olfactometry study. This could explain why no significant differences were observed in our foraging behavior study. Moreover, no differences were found in the emergency rate, which suggests that CABYV-infected plants promote long-term aphid feeding, enhancing virus acquisition from infected source plants and facilitating virus spread in the long term. Thus, parasitoids evolving on aphids feeding on CABYV-infected plants will facilitate their development, as opposed to what happens to those developing on aphids feeding on CMV-infected plants [24]. However, *A. colemani* is able to differentiate among genotypes of the same plant species, indicating the fine-scale differentiation that parasitoids are able to employ in seeking host habitats [44]. In our study, in susceptible melon C311, females preferred air over CABYV-infected plants, and this shows that this response is not a result of HIPVs detection but volatile rejection from an infected plant.

It is known that aphid behavior varies when they develop on plants infected with viruses either transmitted in a non-circulative or circulative manner. Such changes in behavior often facilitate virus spread. In the present study, we have shown that parasitoid behavior also changes when plants are infected with viruses that differ in their transmission mode. Virus infection can reduce plant quality and promote rapid aphid dispersal after probing (e.g., CMV-non-persistent) [14,53] or improve host plant quality for vectors and promote long-term feeding (e.g., CABYV-persistent) [9,14,54]. On the other hand, when aphid vectors develop on plants infected with persistently transmitted viruses in the [9] presence of natural enemies, the feeding duration and vector dispersal can be reduced, thus limiting virus spread from plant to plant [19,23]. The results of our olfactometry assays suggest the following hypothesis: VOCs emitted by persistently transmitted viruses are rejected by female parasitoids, interfering with their host-seeking behavior and limiting their ability to find their aphid host. Such behavioral changes would reduce parasitoid efficacy, which, in turn, would favor virus spread. However, our host-seeking behavior experiment under greenhouse conditions, which was closer to a real crop set-up, showed no significant differences in the parasitism rate between non-infected and CABYV-infected plants, although the parasitism rate was slightly lower on CABYV-infected plants. These results may suggest that *A. colemani* may be disrupted by volatiles emitted by CABYV-infected plants but that other external factors, such as visual cues, may have a higher impact on parasitoid host-seeking behavior. However, further work under more realistic conditions in the field should be conducted to confirm our results. In general conclusion, the transmission of non-persistent and persistent viruses when plants become single and mixed-infected in the presence or absence of aphids and their natural enemies compromises the attraction or rejection of its parasitoids as a long evolutionary process to enhance viral transmissions. Thus, similarly to previous studies that prove virus manipulation of aphid behavior, the results of our olfactory tests support the hypothesis that persistent viruses can also manipulate the behavior of aphid natural enemies in a way that virus transmission is enhanced. Our results also support that in the presence of parasitoids, such virus manipulation is amplified and plant virus infection influences the bottom-up regulation of a plant–aphid–parasitoid system [29].

### 4.3. Impact on Biological Control

Infection of melon plants by CABYV and CMV in simple and mixed infections may have an impact on the efficacy of biological control programs, as these viruses can manipulate the behavior and performance of the major aphid vector *A. gossypii* by improving or decreasing host-plant quality for aphids. Such changes may be differ depending on the mode of virus transmission and our work shows that they could also may influence parasitoid behavior. Our study suggest that biocontrol will be less effective when melon plants are infected with either CABYV or CMV because of the following reasons: (1) When parasitoids are present, they will not be attracted to CABYV-infected plants, thus, aphid population increase and will not be limited because parasitism rate will not be enhanced (functional response in this case will not increase); (2) When plants are infected with CMV, in single or mixed infections, non-infected plants are equally attractive than infected plants to parasitoids, but aphids will prefer to colonize non-infected plants because infected plants are poor hosts for *A. gossypii*. Thus, aphids will likely leave virus-infected host plants before they are parasitized, and CMV spread will not be limited by parasitoids.

## Figures and Tables

**Figure 1 insects-15-00092-f001:**
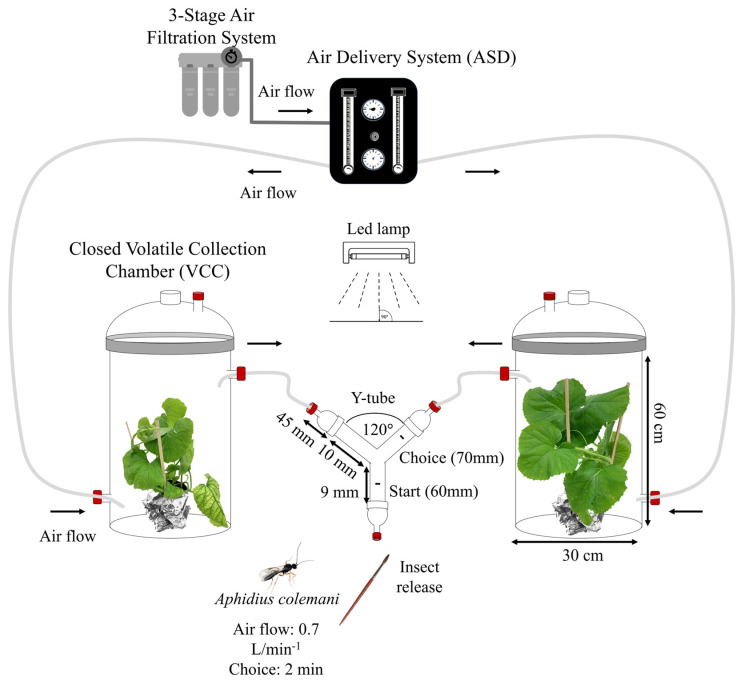
The olfactometer design, known as a Y−tube olfactometer, was employed to evaluate the behavioral responses of both male and female *A. colemani* parasitoids. The airflow was purified using activated charcoal and controlled with a flow meter set to 0.7 L/min. Parasitoids were deemed to have made a choice when they positioned themselves 70 mm above the bifurcation for a duration of 2 min. Each test involved 10 individuals of *A. colemani* and 2 pairs of plants for each set of odor sources.

**Figure 2 insects-15-00092-f002:**
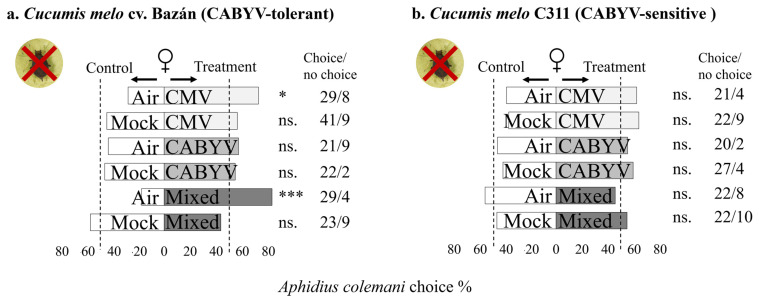
The figure represents the olfactory response of *A. colemani* females to virus-infected melon plants compared to healthy melon plants in two different melon genotypes in the absence of an aphid host. The results are summarized in the bar graphs, with bars indicating the choices made by the parasitoids in different treatment situations. Asterisks (*) denote statistical significance (Significance codes: 0 ≤ “***” < 0.001 < “**” < 0.01 < “*” < 0.05 < “.” < 0.1 < “n.s.” < 1) based on Chi-squared (χ^2^) tests. The “no choice” column represents individuals that did not make a choice within the designated time.

**Figure 3 insects-15-00092-f003:**
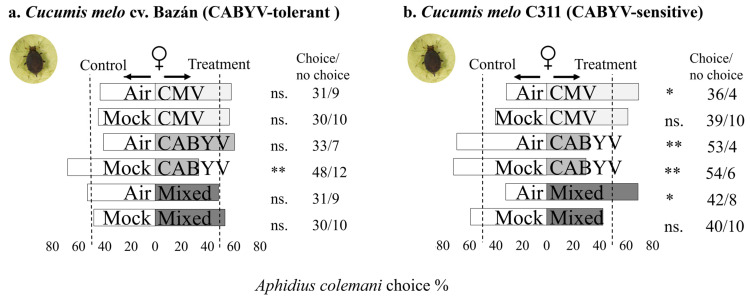
The figure represents the olfactory response of *A. colemani* females to virus-infected melon plants compared to healthy melon plants in two different melon genotypes in the presence of an aphid host. The results are summarized in the bar graphs, with bars indicating the choices made by the parasitoids in different treatment situations. Asterisks (*) denote statistical significance (Significance codes: 0 ≤ “***” < 0.001 < “**” < 0.01 < “*” < 0.05 < “.” < 0.1 < “n.s.” < 1) based on Chi-squared (χ^2^) tests. The “no choice” column represents individuals that did not make a choice within the designated time.

**Figure 4 insects-15-00092-f004:**
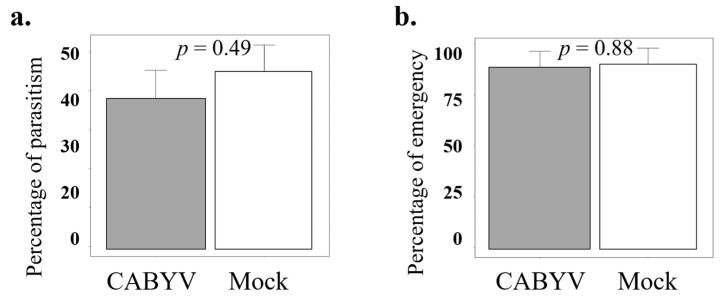
The figures represent the parasitism rate (**a**) and the emergence rate (**b**) of *A. colemani* when exposed to C311 melon plants (sensitive to CABYV) that were either mock-inoculated or infested with at least one hundred second- or third-instar nymphs of *A. gossypii*. The Student’s *t*-test results indicate that there were no significant differences in either parasitism rate or emergence rate between the mock-inoculated and CABYV-infected plants in the semi-natural foraging assay.

## Data Availability

The data that support the findings of this study are available from the corresponding author upon reasonable request.

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
