# Peer review of "Aphidius colemani Behavior Changes Depending on Volatile Organic Compounds Emitted by Plants Infected with Viruses with Different Modes of Transmission"

_insects, 2024, doi:10.3390/insects15020092_

Round 1

Reviewer 1 Report

Comments and Suggestions for Authors

The manuscript entitled “Volatiles emitted by virus-infected melon can disrupt parasitoid attraction to aphid-infested plants” reports that aphid parasitoids react to volatile organic compounds (VOC) emitted by aphid-free and aphid-infested melon plants infected with cucumber mosaic virus (CMV), cucurbit aphid borne yellowing virus (CABYV) or a mixture of both. The manuscript could broadly our comprehensive on the multitrophic interaction in virus-aphid-plant-aphid parasitoid. However, the authors should improve English writing, and the manuscript needs major adjustments. I do not think this current manuscript is suitable for this journal.

Major comments:

1.     I do not understand the significant of two genotypes melon plants used in this study. In fig.3, female parasitoids showed preference for mock-inoculated over CABYV-infected melon in both genotypes, it means CABYV disrupt parasitoid attraction to aphid-infested plants, but the preference of female Aphidius colemani seems not relate with genotypes. On the other hand, in fig.3b, In the case of C311 melon plants, female parasitoids preferred CMV-infected and mixed infected-melon over clean air, but it opposite in CABYV-infected melon plants, whether we can assume that the attraction of CMV-infected plants is stronger than the repulsion role of CABYV-infected melon?

2.     The manuscript used two aphid-borne virus species that are transmitted in non-persistent (NP) and persistent (P), but it is hard to understand that what the mean in this study? what the conclusion we can get from these results?

3.     Importantly, in results section, it is not clearly described. It shows a lot of data about the preference of aphid parasitoids. However, there is no clear way of describing it, which makes it difficult for the reader to understand it. In short, the authors could describe as follows: in case of C311/ Bazán, describe the different results of parasitoids preference in two genotypes. In case of CMV (NP)/CABYV (P), describe the results of parasitoids preference in two different manners. And other similar descriptions.

Minor comments:

1.     the title of the manuscript is one of the results in this study, it cannot reflect the content of this study. Moreover, “emmitted” should be “emitted”, Spelling errors occur elsewhere in the text

2.     lines 266-267, this title could easily misunderstand for only a viral infestation, not reflect previously infested by aphids.

3.     I understand that the authors have inoculated soap, but what is inoculum amount?

4.     Figs should be replaced with high quality image.

5.     Figs S2-S4 are not cited in the text

6.     Lines 308-309, it is (χ2 =5.76; df =1; p = 0.016), not (χ2 =5.82; df =1; p = 0.015)

7.     Lines 336-337, it looks paradoxical that the figure shows that virus-infested plants attract more Aphidius colemani.

8.     Lines 369-370,” To our knowledge, this is the first study that reports………”, but in lines 380-383, “Recently, some studies have ……….” Thus, is really first report?

9.     In fig S4, Cv. Bazán, treatment: Mock (left) Vs. Mix (right). The phenotype of Mock seems grow better than Mix-infected, It′s really?

Comments on the Quality of English Language

the authors should improve English writing.

Reviewer 2 Report

Comments and Suggestions for Authors

The manuscript submitted by Clemente-Orta et al deals about the ability of cucumber mosaic virus of altering the parasitoid attraction to plants infested by Aphis gossypii using specific volatiles. The reported data are very interesting from both an applicative and an ecological/evolutionary point of view. Interestingly, mosaic viruses are able to manipulate not only the behavior of its aphid vector, but also the parasitoid behavior at their own benefit.

The manuscript is generally properly structured and well written. Methods have been adequately described and results are generally well reported and discussed. On the contrary, figures (all of them), even if properly structured, are at low resolution so that in some cases it is difficult to understand the text within them so that they have to be improved.

Requested revisions:

1. Differently from the simple summary that is highly informative and well written,  the abstract has to be improved in order to be more useful in summarizing the main results.

2. statistical significance has to be reported in figure 4. At the same time it could be useful to have figure and caption in the same page and revise the term "percentaje" in both panels;

3. figure 5, even if well structured, could be omitted from the text and used as a graphical abstract (improving its resolution!) since it is not actually useful and the text is sufficiently clear to report what has been observed;

4. the last paragraph is about future studies, but actually Authors included references that can suggest that these topics have been already faced in other published papers. I suggest to remove these last citations;

5. Even if I can understand the choice of using abbreviations, I suggest to use the term persistent  and non-persistent in place of P and NP to help readers. Indeed the term P virus could generate some misunderstanding since, for instance, P virus in Drosophila melanogaster are picornavirus. The same is true for the use of NE in place of natural enemies, please avoid abbreviations that are not really useful.

A last comment is related to the line/section/references spacing that is not in the standard Insects template, but this could be revised at the proofs check.

Comments on the Quality of English Language

The manuscript is generally properly structured and well written. 

Round 2

Reviewer 1 Report

Comments and Suggestions for Authors

Authors have addressed my previous comments, and I have no additional comments.